# Sport Ability during Walking Age in Clubfoot-Affected Children after Ponseti Method: A Case-Series Study

**DOI:** 10.3390/children8030181

**Published:** 2021-03-01

**Authors:** Vito Pavone, Andrea Vescio, Alessia Caldaci, Annalisa Culmone, Marco Sapienza, Mattia Rabito, Federico Canavese, Gianluca Testa

**Affiliations:** 1Department of General Surgery and Medical Surgical Specialties, Section of Orthopaedics and Traumatology, University Hospital Policlinico “Rodolico-San Marco”, University of Catania, 95123 Catania, Italy; andreavescio88@gmail.com (A.V.); alessia.c.92@hotmail.it (A.C.); annalisa.culmone@libero.it (A.C.); marcosapienza09@yahoo.it (M.S.); mattia.rabito@gmail.com (M.R.); gianpavel@hotmail.com (G.T.); 2Department of Pediatric Orthopedic Surgery, Jeanne de Flandre Hospital, Lille University Centre, 59000 Lille, France; canavese_federico@yahoo.fr

**Keywords:** clubfoot, CTEV, sport, sport practice, sport activity level, young athletes, ponseti method

## Abstract

Background: The Ponseti method (PM) of manipulative treatment for congenital talipes equinovarus (CTEV) or clubfoot became widely adopted by pediatric orthopedic surgeons at the beginning of the mid-1990s with reports of long-term successful outcomes. Sports are crucial for children’s development and for learning good behavior. This study aimed to evaluate the sports activity levels in children treated with PM and to assess the different outcomes, according to gender and bilaterality. Methods: A total of 25 patients (44 feet) with CTEV treated by the PM were included in the study. The patients were clinically evaluated according to the Clubfoot Assessment Protocol, American Orthopedic Foot and Ankle Society, Ankle–Hindfoot score, the Foot and Ankle Disability Index (CAP, AOFAS, and FADI, respectively), and FADI Sport scores. Results: The overall mean CAP, AOFAS, FADI, and FADI Sport scores were 97.5 ± 6.4 (range 68.75–100), 97.5 ± 5.8 (range 73.00–100), 99.9 ± 0.6 (range 97.1–100), and 100, respectively. Gender and bilaterality did not affect outcome (*p* > 0.05). Conclusions: The data confirmed good-to-excellent outcomes in children with CTEV managed by PM. No limitations in sport performance or activity could be observed. In particular, male and female patients and patients with unilateral or bilateral involvement performed equally well.

## 1. Introduction

Congenital talipes equinovarus (CTEV) is one of the most common congenital pediatric orthopedic deformities and is characterized by dorsal hyperflexion of the foot, varus of the hindfoot, forefoot adduction and increased plantar arch [1]. Clinical manifestations may depend on etiology [1], severity, and clinical course [2,3], and different treatment options are available to treat patients with CTEV [4,5,6,7,8]. 

The Ponseti method (PM) of manipulative treatment for CTEV became widely adopted by pediatric orthopedic surgeons beginning in the mid-1990s, with reports of long-term successful outcomes [9,10]. PM consists of a series of specific manipulations and cast applications to concurrently correct the forefoot, midfoot, and subtalar components of the deformity; a percutaneous Achilles tenotomy is often needed to correct the equinus component. Correction is then maintained for the first few years using a foot abduction orthosis at night and during naps. The aim of the procedure is to achieve a pain-free supple plantigrade foot with a minimal amount of surgery as practicably possible as long-term studies on the outcomes of surgical releases have reported high rates of painful and stiff feet with poor post-surgical functional outcomes [6,7,8,9,10]. 

Sport activities are crucial for children’s development and for learning good behavior [11,12,13,14,15]. During childhood, sports may prevent future pathologies, and these activities are essential for the social inclusion and psychological well-being of the child [16,17,18]. Moreover, it has been shown that young sport practitioners have improved quality of life [11], brain cortical excitability [12], long-term neural adaptation mechanisms, and visuo-spatial capacities [14]. 

Compared to other techniques, the PM has been shown to preserve motor activities and to allow almost normal motor gross function development of children with CTEV [19,20]. In particular, Debra et al. [21] have shown a minimal delay (1.5 months) in gross motor milestone achievement, and Lohle-Akkersdijk et al. [22] reported a slight decrease in the walking speed of children with CTEV. Additionally, some authors have found that children with bilateral CTEV have worse balance and body coordination than patients with a unilateral deformity [20,23]. A few studies have assessed the sports abilities in CTEV-affected patients treated with different procedures, and PM was noted as the most effective in preserving good sports performances [10,23].

The main objective of this study was to evaluate the sports activity levels in children with CTEV managed by PM. The secondary aim was to evaluate whether any differences in sport activity performance exist between male and female patients with unilateral or bilateral CTEV; it was hypothesized that children with CTEV treated by PM have good functional outcomes and good sports activity performances regardless of gender and bilaterality.

## 2. Materials and Methods

### 2.1. Sample Eligibility Criteria

Between 2010 and 2020, 79 children with CTEV were treated with PM at the Section of Orthopaedics and Traumatology, University Hospital Policlinico “Rodolico-San Marco”, University of Catania, Catania. 

Inclusion criteria consisted of several requirements: (1) confirmed diagnosis of idiopathic CTEV (Forefoot adductus, midfoot cavus, hindfoot varus, hindfoot equinus); (2) initial treatment according to the PM; (3) patients > 3 years of age; (4) sports activities at a recreational or occasional level; and (5) > 2 years of follow-up.

Exclusion criteria consisted of several parameters: (1) non-idiopathic CTEV; (2) patients with underlying neurological or neuromuscular condition; (3) patients < 3 years of age; (4) no participation in sports activities; and (5) follow-up < 2 years.

According to inclusion and exclusion criteria, 36 patients were considered eligible for the study, while 11 patients were lost to follow-up and were excluded from the analysis (30.6%). A total of 44 CTEV in 25 patients were retrospectively reviewed and included in the present study. 

All patients provided an informed consent to participate in the present investigation. This study was carried out according to the guidelines for Good Clinical Practice and the Declaration of Helsinki.

### 2.2. Ponseti Method Treatment Protocol

PM [7] consists of weekly sessions of manipulations and long-leg plaster casts with the knee at 90° of flexion. The methods onset has been described since the first weeks of life [4]. The manipulations should abduct the forefoot around the talus after the latter has been stabilized as well as mobilize the feet gently in all planes and should regularly stimulate lateral peroneal muscles in order to prevent internal rotation [4]. The first step of the cast treatment is to correct the cavus by supinating the forefoot to restore the correct arch of the foot. Subsequently, the foot is progressively abducted around the talus. Last, the equinus is corrected by dorsiflexing the foot. If the dorsiflexion of the ankle remains below 10°, percutaneous Achilles’ tenotomy is performed in the operating room under general anesthesia. Following Achilles’ tenotomy, a long leg cast is applied for 3 weeks, with feet externally rotated and knee flexed at 90°; after cast removal, feet are placed in splints with 60° to 70° of external rotation. The splints are worn 23 hours a day during the 3 month period, after which time they are only worn during naps and at night until age five years. Clinical follow-up is performed every six months until six years of age.

### 2.3. Clinical Assessment

Clinical and functional outcome were evaluated in patients with at least 2 years of follow-up by using the Clubfoot Assessment Protocol [24], the American Orthopedic Foot and Ankle Society Ankle–Hindfoot score [25], the Foot & Ankle Disability Index (CAP, AOFAS, and FADI, respectively) and FADI Sport scores [26]. 

The CAP contains 22 items divided into four sub-groups: (1) mobility (eight items), (2) muscle function (three items), (3) morphology (four items), and (5) motion quality I and II (seven items). The scoring is divided systematically in proportion to what is regarded as normal variation and its supposed impact on perceived physical function ranging from 0 (severe reduction/no capacity) to 4 (normal). Score grading can vary between three and five levels, and it can be used with sufficient reliability during the first seven years of childhood (it is age-independent) by examiners with good clinical experience [24]. 

The AOFAS Ankle-Hindfoot score consists of nine items under three different categories: (1) pain (40 points), (2) functional aspect (50 points), and (3) alignment (10 points), totaling 100 points. Items on pain and functional limitation are answered by the patient, while the alignment items are answered by an examiner [25]. 

The FADI is a region-specific self-reporting scale of function that includes 34 items divided into two subscales: (1) the first (FADI) consists of 26 items about activities of daily living (ADL) and pain and (2) the second (FADI Sport) consists of eight items about sports activities. Higher scores represent higher levels of function for each subscale. The ADL and sports subscales are scored separately [26]. 

Clinical assessment data were collected and analyzed by two authors.

### 2.4. Statistical Analysis 

Continuous data are presented as means and standard deviations as appropriate. The Student’s *t*-test was used to evaluate the mean and standard deviation between subgroups. Chi-square tests were used to verify the homogeneity of the group. Pearson’s correlation coefficient was used to assess the correlation between the clinical scores and CTEV severity according to Pirani Score or the cast number.

The selected threshold for statistical significance was *p* < 0.05. All statistical analyses were performed using the 2016 GraphPad Software (GraphPad Inc., San Diego, CA, USA). 

## 3. Results

### 3.1. Sample

The cohort of 25 patients (44 feet) consisted of 19 male (76%) and six female (24%) patients, and the mean age at time of evaluation was 6.4 ± 2.5 years (range 3–12). Eighteen patients had a bilateral (75%), and six unilateral CTEV (three left and three right). 

The mean age at start of treatment was 17.2 ± 10.7 days (range 14–48). All patients had clinical follow-up for at least two years (mean: 4.6 ± 2.4 years; range 2.9–11.8). The mean number of casts per patient was 6.7 ± 1.9 (range 5–10) and 36 out of 44 feet underwent percutaneous Achilles tenotomy under general anesthesia (82%). The mean activities of daily living (ADL) Pirani’s score at the beginning of treatment was 4.9 ± 1.0 (range 3–6) for included patients. If orthopedic treatment was ineffective, and feet showed no improvement, further surgery was performed. Overall, tibialis anterior transfer was performed in 1/44 feet (2.3%); no cases of posterior or medial release were recorded. Sports in which children participated included fitness (14 cases; 56.0%), soccer (four; 16%), swimming (four; 16.0%), and other activities (three; 12%), as shown in Table 1.

### 3.2. Clinical Assessment

According to the CAP questionnaire, the mean recorded score was 97.5 ± 6.4 (range 68.75–100). Similarly, the average AOFAS was 97.5 ± 5.8 (range 73.00–100), and the average FADI score was 99.9 ± 0.6 (range 97.1–100). The average FADI Sport score for the whole cohort of patients was 100 (Table 2). 

### 3.3. Clinical Assessment: Gender Comparison

Male and female patients had mean CAP scores of 97.2 ± 6.9 and 99.2 ± 1.6, respectively (*p* = 0.49). Similarly, no statistically significant differences could be recorded with the AOFAS score), with mean values of 97.1 ± 6.5 and 99.2 ± 1.9 for males and females, respectively (*p* = 0.45). The mean FADI score was 99.8 ± 0.7 and 100 in males and females, respectively (*p* = 0.5). The mean FADI Sport score of the whole cohort was 100 (*p* = 1), as shown in Table 2. 

### 3.4. Clinical Assessment: Side Comparison

No statistically significant differences could be identified in the mean CAP score of unilateral clubfoot (99.6 ± 0.6) versus bilateral (97.2 ± 6.9) CTEV (*p* = 0.41). According to the AOFAS score, the mean was 100 and 97.1 ± 6.2 for unilateral and bilateral involvement, respectively (*p* = 0.27). The average FADI score was 100 ± 0.0 and 99.8 ± 0.7 in unilateral and bilateral cases, respectively (*p* = 0.5). Both groups reported a mean FADI Sport score of 100 (*p* = 1), as shown in Table 2.

### 3.5. Clinical Assessment and Correlations

No statistical correlation between the scores and CTEV severity according to the initial Pirani Score (*p* > 0.05) was found. Between the scores and applied cast numbers, no statistical correlation (*p* > 0.05) was recorded. The correlations data are reported in Table 3.

## 4. Discussion

Patients with CTEV treated with the PM have shown excellent functional outcome at mid- and long-term evaluation periods. At the same time, good-to-excellent sports performances were found among the included patients. In particular, no differences in functional outcome or sports activity performances could be identified when comparing male and female subjects or patients with unilateral and bilateral CTEV. Several articles have reported good outcome in patients with CTEV managed by the PM [6,27,28,29,30]. However, the literature is lacking in studies investigating the sports activity levels in children with CTEV during walking age. 

Recent data highlight that patients with CTEV managed by the PM had better sports performances when compared with patients managed by other techniques [10,23]. In 2017, a clinical trial compared the sports performances of children treated by Bösch method, the Cincinnati procedure, and PM. The authors found that children managed by the PM experienced less difficulty and less pain when participating in sports compared to patients treated with other methods [10]. In addition, according to three-dimensional (3D) gait analysis, patients treated by PM had better running speed/agility (*P* = 0.019), body coordination (*p* = 0.038), and strength (*p* = 0.007) compared to patients treated with the French Functional Physical Therapy method [23]. 

Two studies compared the athletic abilities of children with CTEV, and compared them with the general population [31,32]. Kenmoku et al. [31] assessed 30 children with CTEV treated with PM at a mean age of 9.2 ± 1.9 years (range 7–12); all patients had an excellent Ponseti functional score and any difference between individuals with CTEV and the general population could be identified for the 50 meter run, standing long jump, 20-m shuttle meter run, repetition side steps, and sit-ups. The mere inconsistency highlighted by the authors is a different podalic pattern of the center of pressure during the walk and the running. Mir et al. [32] prospectively evaluated 48 children with CTEV using the Roye Disease Specific Index and the Physical Activity Questionnaire-Elementary School (PAC-ES) adjusted to the Irish population. Almost all included patients (97%) were able to participate in school-based physical education activities. Furthermore, all patients participated in extra-curricular sporting activities; 80% participated with a frequency of 4 to 7 days per week compared to 17% of the general population. The authors concluded that sporting participation of patients with idiopathic CTEV managed by the PM was excellent, especially for extracurricular activities. 

Our study examined 25 patients with a mean age of 6.4 ± 2.5 years (range 3–12) at the time of the evaluation; the whole sample presented excellent results according to specific CTEV evaluation and sport activities in addition to specific foot and ankle assessments. The cohort highlights a good correction of the foot, and only one patient required additional surgery (tibialis anterior transfer; 2.3%) although sport participation was not negatively affected. To evaluate the sports abilities and performances of the children, the FADI Sport, a specific questionnaire consisting of eight items about sports activities, was chosen. The score assesses running, landing, lateral movements, ability to perform activities with normal technique, jumping, squatting and stopping quickly, low-impact activities, and ability to participate in desired sports with no limitations. In each of these items, the maximum rating was observed. 

Mir et al. [32] reported outcomes of 16 and 19 patients with bilateral and unilateral CTEV, respectively, but did not compare these two subgroups. To the best of our knowledge, our study is the first one to investigate possible athletic ability differences according to gender and bilaterality. No differences were found in the comparison between the different subgroups. Males and females had excellent functional outcomes and could perform the preferred activity without any limitations. Similarly, patients with unilateral and bilateral CTEV have similar sport performances despite the reduced size of the affected calf [33]. Moreover, no correlations were found between the clinical scores and CTEV severity according to Pirani Score or the cast number.

Debra et al. [21] showed a minimal delay (1.5 months) in gross motor milestone achievement, and Lohle-Akkersdijk et al. [22] reported a slight decrease in walking speed of children with CTEV. Additionally, some authors found that children with bilateral CTEV have worse balance and body coordination than patients with unilateral deformities [20,23]. It could be hypothesized that these findings are related to the use of braces/splints during the early years of life. During the growing age, the progressive reduction of brace/splint time per day and the progressive involvement of children in different school, social, and sports activities may contribute to improved functional and athletic performance. 

Several limitations in the analysis of our results can be described. First, this study was a single-center retrospective study, the number of patients was relatively small, and no control group was used. However, we are able to offer some evidence on outcomes of sport performance of children with CTEV treated by the PM. Moreover, patients were not followed until skeletal maturity; thus, some feet issues could potentially recur, and some patients may become symptomatic with a subsequent decrease in sports performances. Therefore, it is possible that a longer multicenter follow-up study might be necessary to predict the long-term outcome of this specific treatment option.

## 5. Conclusions

In conclusion, the data confirmed good-to-excellent outcomes in children with CTEV managed by PM. No limitations in sports performances or activities could be observed. In particular, male and female patients, and patients with unilateral or bilateral involvement, perform equally well.

## Figures and Tables

**Table 1 children-08-00181-t001:** Group demographics.

Group	Patients	Gender		Mean Age (Years)
		M	F	
Sample	25	19	6	6.4 ± 2.5
Unilateral	6	4	2	6.5 ± 2.3
Bilateral	19	15	4	6.4 ± 2.6

M = male; F = female.

**Table 2 children-08-00181-t002:** Clinical assessment. Results reported in mean and standard deviation.

Group	Patients	CAP	AOFAS	FADI	FADI Sport
Sample	25	97.5 ± 6.4	97.5 ± 5.8	99.9 ± 0.6	100 ± 0.0
Unilateral	6	99.6 ± 0.6	100 ± 0.0	100 ± 0.0	100 ± 0.0
Bilateral	19	97.2 ± 6.9	97.1 ± 6.2	99.8 ± 0.7	100 ± 0.0
p Uni vs Bil		**0.41**	**0.27**	**0.50**	**1.00**
Male	19	97.2 ± 6.9	97.1 ± 6.5	99.8 ± 0.7	100 ± 0.0
Female	6	99.2 ± 1.6	99.2 ± 1.9	100 ± 0.0	100 ± 0.0
p Male vs Fem		**0.49**	**0.45**	**0.50**	**1.00**

Uni = Unilateral; Bil = Bilateral; CAP = Clubfoot Assessment Protocol; AOFAS = American Orthopedic Foot and Ankle Society (AOFAS) Ankle-Hindfoot score; FADI = Foot & Ankle Disability Index.

**Table 3 children-08-00181-t003:** Clinical assessment and correlations.

Correlation PCC (95% CI)	CAP	AOFAS	FADI	FADI Sport
Clinical scores and Pirani Score	−0.07 (−0.27; 0.14)	−0.05 (−0.26; 0.16)	−0.10 (−0.30; 0.11)	0.00 (−0.27; 0.27)
Clinical scores and Casts Number	−0.07 (−0.14; 0.27)	0.05 (−0.16; 0.26)	−0.02 (−0.38; 0.02)	0.00 (−0.27; 0.27)

PCC = Pearson Correlation Coefficient; CI = confidence interval; Uni = Unilateral; Bil = Bilateral; CAP = Clubfoot Assessment Protocol; AOFAS = American Orthopedic Foot and Ankle Society (AOFAS) Ankle-Hindfoot score; FADI = Foot & Ankle Disability Index.

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
