# Peer review of "Sport Ability during Walking Age in Clubfoot-Affected Children after Ponseti Method: A Case-Series Study"

_children, 2021, doi:10.3390/children8030181_

Round 1

Reviewer 1 Report

  1. Introduction correctly described problems
    Perhaps it is worth emphasizing the innovativeness of scientific research, treating the second goal as the basic one.
  2. Materials and Methods
    a. "Our Instition" - in which department the procedures were performed (two were given in affiliation)
    b. The method of treatment described is correct including the percentage of patients treated with tenotomy.
    c. Indeed, the study group may have to be more numerous. 
    d. There is no control / comparison group
    e. The relevant parameters for the functional assessment were taken into account, but it was not stated by who made the functional assessment (one or more people).
    f. Over 2 years of follow-up - it may be worth repeating the assessment of patients after another 5 years
  3. Results presented and described correctly
  4. Discussion
    Site 5: 2 paragraph refers to the comparison of other treatments that were not considered in this work and that was not the goal of treatment. Maybe move this fragment to the introduction?
    Site 6: second acapit link to [34] is not in the reference list (I think it was supposed to be [33])

Author Response

Reviewer 1

  1. Q1) Introduction correctly described problems
    Perhaps it is worth emphasizing the innovativeness of scientific research, treating the second goal as the basic one.

A1) Thanks for your positive comment.

  1. Materials and Methods
    Q2) a. "Our Instition" - in which department the procedures were performed (two were given in affiliation)

A2) The institution name was included in the manuscript.
Q3) b. The method of treatment described is correct including the percentage of patients treated with tenotomy.

A3) Thanks for your positive comment.
Q4) c. Indeed, the study group may have to be more numerous. 

  1. There is no control / comparison group

A4) We agree with your suggestion. Ponseti method requests a close follow-up until age of 5-6 years. Long follow-up studies often record an high patients drop-out and the literature is poor of long- term studies concerning the clubfoot. Moreover, in our study the age ranges between 3 and 12 years with a mean follow up of 4.6 years. Unfortunately, it was not possible include a larger number of patients. The small size of the sample is included among the limits of the study. At the same time, we agree with the reviewer about the group control lack reduce the scientific relevance of the paper. In order to solve the limit subgroups comparisons were performed.

Q5) e. The relevant parameters for the functional assessment were taken into account, but it was not stated by who made the functional assessment (one or more people).

A5) The requested information was added in the text

Q6) f. Over 2 years of follow-up - it may be worth repeating the assessment of patients after another 5 years

A6) Thanks for your positive suggestion. In the present study the sample were evaluated once. We would like to include more follow-ups in future trials.

  1. Q7) Results presented and described correctly

A7) Thanks for your positive comment.

  1. Discussion
    Q8) Site 5: 2 paragraph refers to the comparison of other treatments that were not considered in this work and that was not the goal of treatment. Maybe move this fragment to the introduction?

A8) Additional information was added in introduction
Q9) Site 6: second acapit link to [34] is not in the reference list (I think it was supposed to be [33])

  1. A9) Thanks for your suggestion. the typo was corrected

Reviewer 2 Report

Thank you for the opportunity to review this paper which I thoroughly enjoyed reading. This paper evaluates the Ponseti method (PM) of manipulative treatment for CTEV became widely adopted by pediatric orthopedic surgeons beginning in the mid-1990s and long-term successful outcomes have been reported. The sport activity is crucial for children development and behaviors. The manuscript is excellently written, with clear research aims and methodology. The statistical methods are sound and accurate and the discussion, limitations and conclusions are relevant to the results. Furthermore, in my opinion the topic and premise of the study would sit well within the journal to which it was submitted. I have no major recommendations for the content. These will be discussed below relative to the sections of the manuscript. 
 1) Title: The title is correct as it reflects correctly the objective and hypothesis of the work.
2) Summary:  This section follow a well structured format.
3) Introduction:  The research question itself is sound and the topic is strongly introduced. 
4) Materials and Methods: The inclusion and exclusion criteria appear sound.  
5) Results: The results is clear and concise with appropriate statistical analysis been performed appropriately and rigorously.
6) Discussion: The discussion appears well developed and appropriate, authors describe the results, the limitations and compare with other researchs. 
7) Conclusion: The conclusion is conclusively.
8) References: Appropriate
9) Tables: Correct.

Accept in its initial format

Author Response

We are grateful to the reviewer for the positive comments.

Reviewer 3 Report

INTRODUCTION

It´s needed to complete this sentence at final goals part:

The main objective of this study is to evaluate the sport activity level in children with CTVE managed by the PM…….. compared to/with…?

The secondary goal is not supported on the introduction. It´s needed to explain it, with proper bibliography references.

MATERIAL AND METHODS

Sample

“…at our Institution….” What institution?

Inclusion criteria: its needed  to support the criteria referred to diagnosis and characteristics of CTVE

Ponseti Method

“…to correct the cavus by supinating the forefoot…” please, explain the concept of supinating de forefoot to correct the cavus arch, if the forefoot is not the arch and, in addition,  if this kind of feet already have supinating the forefoot with their  adduction and plantarflexion

What is the ideal age to perform this method? You have to write it on this section.

“…The splints are worn 23 hours a day during the 3,…”  3 …what? Days? Months? Weeks?

DISCUSION

It don’t show the proper view of the article, which is focused on sport activity comparison between CTVE treated with or without PM, and there is any reference to sport activities of the participants, but use CAP and others evaluation club foot scales.

CONCLUSIONS

The conclusions are not supported by the results neither methods achieved on the present study.

 Author Response

INTRODUCTION

Q1) It´s needed to complete this sentence at final goals part:

The main objective of this study is to evaluate the sport activity level in children with CTVE managed by the PM…….. compared to/with…?

A1) Unfortunately no comparisons were performed in the main aim. For this reason we believe that the purpose of the study is properly explained.

Q2) The secondary goal is not supported on the introduction. It´s needed to explain it, with proper bibliography references.

MATERIAL AND METHODS

Sample

Q3) “…at our Institution….” What institution?

A3) The institution name was included in the manuscript.

Q4) Inclusion criteria: its needed  to support the criteria referred to diagnosis and characteristics of CTVE

A4) the criteria were better explained in methods.

Ponseti Method

Q5) “…to correct the cavus by supinating the forefoot…” please, explain the concept of supinating de forefoot to correct the cavus arch, if the forefoot is not the arch and, in addition,  if this kind of feet already have supinating the forefoot with their  adduction and plantarflexion

A5) Thanks for your comment. Medial arch lies between the head of first metatarsus and calcaneus. A gentle pression on the plantar surface of the head of first metatarsus reduces the medial arch height and causes the forefoot supoination.

Q6) What is the ideal age to perform this method? You have to write it on this section.

A6) Additional clarifications were added.

Q7) “…The splints are worn 23 hours a day during the 3,…”  3 …what? Days? Months? Weeks?

A7) The word “months” was added.

DISCUSION

Q8) It don’t show the proper view of the article, which is focused on sport activity comparison between CTVE treated with or without PM, and there is any reference to sport activities of the participants, but use CAP and others evaluation club foot scales.in

A8) the literature is poor of study aimed to assess the sport ability in CTEV affected patients. In the discussion, the principal findings were reported. Moreover, often the sport abilities were assessed according to different scales. For these reasons the comparison are difficult.

CONCLUSIONS

Q9) The conclusions are not supported by the results neither methods achieved on the present study.

A9) In conclusion section, we reported our findings according to the study primary and secondary aims

Round 2

Reviewer 1 Report

Accept in present form

Reviewer 3 Report

Thank you very much for clarify the request